# *Sambandha* as a '*Śakti*-of-*Śakti*s': Bhartṛhari's Influence on the Relational Realism of Pratyabhijñā

## Jesse Berger

Divinity School, University of Chicago, Chicago, IL 60637, USA; jesse727@uchicago.edu

**Abstract:** Contemporary scholarship has significantly advanced our understanding of the grammarian Bhartṛhari's influence on the Pratyabhijñā Śaivism of Utpaladeva and Abhinavagupta. One area that has been somewhat neglected, however, is the subject of relation (*sambandha*). Here, I examine the influence of Bhartṛhari's *sambandha-vāda* on the Pratyabhijñā school. As I see it, Bhartṛhari's understanding of the holistic movement of *sphoṭa*—the practical process of 'encoding' and 'decoding' linguistic information—leads to a necessary reevaluation of the *general* logical form of *sambandha*, i.e., 'relationality-as-such.' On this account, Bhartṛhari articulates a basically *transcendental* conception of *sambandha* as a '*śakti-of-śakti*s' in his '*Exposition of Relation*' (*Sambandhasamuddeśa* [SSam]). This effectively means that one cannot designate the general logical form of *sambandha* in linguistic terms without also thereby changing its essential nature as such (cf. Houben: 170–4). I maintain that Utpaladeva's '*Proof of Relation*' (*Sambandhasiddhi* [SS]) leverages this insight into a series of pragmatic arguments to demonstrate that *vimarśa*, or recognitive judgment, is the true locus of relational action—i.e., unity-in-diversity (*bhedābheda*). In doing so, he effectively salvages a coherent understanding of relation as *necessarily* real (*satya*) from the deconstructive agenda of the Buddhist eliminativist, even though the referent may indeed appear paradoxical from the perspective of theoretical reason alone.

**Keywords:** Indian philosophy; relations; Bhartṛhari; Utpaladeva; Pratyabhijñā; transcendental argument; realism; idealism





## 1. Introduction

Bhartṛhari's influence on the Pratyabhijñā thinkers Utpaladeva and Abhinavagupta is now well-established.[1] Many of the grammarian's central concepts are appropriated and developed in their own absolute idealist system of philosophy. Among these are the intrinsic self-consciousness of each cognition (*svasaṃvedana*) (which entails that any single cognition is unable to be the object of another cognition) and his keystone doctrine that all knowledge, including perceptual cognition, is fundamentally linguistic.[2] While there has been great progress with respect to these important topics, the subject of *sambandha* has received relatively little attention from scholars of Pratyabhijñā.[3] To be sure, this is partly because the *Sambandhasiddhi* [SS], Utpaladeva's main treatise on relations, had not been fully translated until very recently.[4] Now, we can better appreciate the ways in which Bhartṛhari's influence also extends to the Pratyabhijñā theory of *sambandha*, the central topic of Utpaladeva's SS, and a recurrent theme in his primary work, the *Īśvarapratyabhijñākārikā* [ĪPK].[5] Although not explicitly mentioned in Utpala's SS (or, for that matter, in the relevant portions of the ĪPK), I will show that the Pratyabhijñā thinkers basically defend and refine Bhartṛhari's analysis of relation in the third portion of his *Vākyapadīya* [VP], the *Sambandhasamuddeśa* [SSam].[6]

The structure of this paper is as follows: Since I contend that Bhartṛhari's theory of *sambandha* results from his monistic metaphysics of divine speech (*para-vāc*), in the second section, I analyze Bhartṛhari's theory of *sphoṭa*, which depicts a progressive, triadic transformation of the holistic essence of undifferentiated *vāc* into determinate sequences of

particular word-sounds (*dhvani-śabda*). In the third section, I show that this 'vertical' realization of practical speech-acts motivates and informs Bhartṛhari's novel interpretation of the general form of *sambandha* as a *śakti-of-śaktis*—i.e., a transcendental capacity of speech that unifies a diversity in the abstract designation of determinate linguistic signs but cannot be expressed or designated in those very terms. Before discussing Utpaladeva's appropriation of this idea in the SS, in the fourth section, I briefly outline the Dharmakīrtian position as presented by Utpala—to wit, that 'relations' do not exist, because nothing ultimately real can instantiate a dyadic form that is simultaneously unified and diverse. On my reading, Utpala emphasizes that the nominalist axioms of Dharmakīrti's ontology necessarily entail that all forms of 'relation' are no different from *universal* 'conceptual constructions' (*kalpanā*)—that is, a false, conventional superimposition (*āropa*) of distributed/continuous (*anvaya*) properties onto the 'ultimate' reality of momentary, non-relational particulars. The fifth and sixth sections turn to the Pratyabhijñā position that leverages Bhartṛhari's coextension of relation with 'a subtle form of *śakti*' (cf. MacCracken 2017) to overcome the eliminativism of Buddhist nominalism. Utpala first argues that relation cannot be a form of conceptual 'error,' because we presuppose a form of relation in the practical judgment thereof. He then offers a set of pragmatic arguments that leverage the coextension of *sambandha* and *śakti* to advance his own Śaivite form of absolute idealism: namely, the true locus of relational action—viz., the *śakti* that manifests (*prakāśa*) all determinate appearances—is nothing other than the general unity-in-diversity of recognitive judgment (*vimarśa*) itself.[7]

## 2. Sphoṭa: Bhartṛhari and the Metaphysics of Language

To understand Bhartṛhari's conception of *sambandha* and its impact on these later Śaiva idealists, we must first briefly look at his theory of meaning, or *sphoṭa*. For although the Pratyabhijñā tradition does not embrace *sphoṭa* semantics explicitly (and Utpala does not seem to elaborate another alternative in the extant works) the Pratyabhijñā tradition *does* adopt Bhartṛhari's theory that *vāk*, or divine speech, manifests through progressive phases of realization.[8] This, I claim, generates a certain theory of the general form of *sambandha*. Yet, in Bhartṛhari's system, these two ideas go hand in hand—it is impossible to talk about the gradations of *vāk* without at least briefly touching upon the nature of *sphoṭa*. Both doctrines, in turn, stand against the backdrop of Bhartṛhari's linguistic monism; everything that exists is constituted by, and proceeds from, a single infinite reality, what he calls '*śabda-brahman*' (literally 'Brahman [in the form of] word/sound').[9] This *śabda-brahman* represents the absolute and unified ground of all worldly phenomena and knowledge.[10] However, quite unlike the 'idealism' of, say, Advaita Vedānta or Yogācāra Buddhism (insofar as we may describe them as such), where the determinate and differentiated conceptual trappings of language reflect the quintessence of *saṃsāric* ignorance, Bhartṛhari believed that salvation comes with the realization that the linguistic nature of thought is identical with the transcendent activity that generates and sustains the conventional world (Matilal 1990, p. 95). Thus, the embodied *capacity* to express reality *linguistically*; the conceptual structures of language *itself*; and the subjective *apprehension* thereof; all are nothing other than the primordial reality of divine speech (*para-vāc*).

*Sphoṭa* is a concept that Bhartṛhari develops from the Vedic grammarians to designate the holistic process of this divine language.[11] While it literally means 'bursting,' 'expansion' or 'disclosure,' for the ancient grammarians, it comes to represent the inner principle of all semantic phenomena, which is manifest in the fact that diverse externalized verbalizations (*śabda*) convey a universal unit of meaning or reference (*artha*).[12] We might call it something like a holistic intuition of pure semantic unity that both *generates*, and yet is also *expressed by*, sequential and distinct word-sounds (*śabda*).[13] Patañjali defines it as 'the eternal and imperceptible element of sounds and words and the real vehicle of the idea which bursts or flashes on the mind when a sound is uttered.'[14] *Sphoṭa* is therefore the part-less, non-sequential, indivisible 'semanticality' of any particular speech-act.[15] In this respect, words

and sentences both have *sphoṭa*, but the *sphoṭa* of sentences is not reducible to the *sphoṭa* of words.[16]

It is therefore important to note that *sphoṭa* is not merely the universal referent (*artha*) of a sentence or word (since both are technically *sphoṭa*) but the holistic 'bursting forth' of a unified idea that occurs in the mind of a particular speaker and hearer, for the phenomenon of language means the disclosure of an innate capacity to recognize that verbalized sounds (*śabda*) are related to a meaning (*artha*). Bhartṛhari writes that 'the wise discern two [aspects] in the expression of words: one [i.e., *sphoṭa*] is the cause of the word [and] the other [*dhvani*] is used to convey meaning.'[17] The term *dhvani* here refers to what we might call the *tokenized* expressions of language. They are the spoken and heard 'articulated sounds' that practically manifest the concealed and implicate unity of *sphoṭa*.[18] As Matilal puts it, 'Language and meaning are not two separate realities such that one *conveys* the other. They are in essence the two sides of the same coin. The *sphoṭa* is the unitary principle where the symbol and what is signified are one' (95). And while Matilal is technically correct, his talk of strict identity of sign and signified occludes, I think, the *triadic* dimension of *sphoṭa*. That is to say, *sphoṭa* actually entails that (i) the vibrations of physical sound, (ii) the inner activity of thinking, and (iii) the determinate appearance of linguistic signs, are all internally related concepts.[19]

To explain the gradual unfolding of this continuous structure of *sphoṭa* into the diversity of *dhvani*, Bhartṛhari postulated three stages of speech: (i) 'articulate speech' (*vaikharī vāc*), or the external, public, tokenized utterances of individual speakers; (ii) 'intermediate speech' (*madhyamā vāc*), which reflects a pre-verbalized form of mental representation;[20] and (iii) 'seeing speech' (*paśyanti vāc*), the innermost realm of indivisible semantic unity, which is non-sequential (*akramā*), undivided (*abheda*) and concealed (*pratisaṃhṛta*).[21] Although *paśyanti vāc* exhibits no determinate content, it serves as a universal plenum for all semantic potentials. Accordingly, every self or mind consists of an infinite storehouse of *sphoṭa*-potentials, the universal types of semantic knowledge withdrawn from tokenized linguistic expression. This is the primordial, eternal realm of language, where universal possibilities are 'seen' ('*paśya*') or conceptualized in the mind without being explicitly verbalized. When a speaker has the practical desire to communicate, they are aware of the corresponding *vākya-sphoṭa* and initiate a formal procedure for its encoding into *vaikharī vāc*. Thereupon, at the level of public discourse, another person hears the individual *pada-sphoṭas* and these trigger a decoding process that results in the hearer's holistic recognition of the very same *vākya-sphoṭa* within the mind of the speaker (Coward 1997, p. 77). Cognitive activity therefore resembles a spontaneous procedure of encoding and decoding the holistic essence of universal language into determinate and particular speech-acts (Seneviratne 2015, p. 140).[22]

Importantly for our purposes, the inclusion of an intermediate level of speech processing (*madhyamā vāc*) suggests that the movement from the indeterminate unity of *paśyanti vāc* into the determinate expressions of *vaikharī vāc* is a *graded realization*. In other words, the linguistic activity that manifests *sphoṭa cannot be an instantaneous transition*. To the extent that we view 'intentionality' as broadly coextensive with discursive activity, this suggests that the capacity to utilize and understand the sequential terms of public language rests on a deeper awareness of *their progressive manifestation*. The general three-fold procedure for 'encoding' and 'decoding' *sphoṭa* thus chiefly reflects stages in a *phenomenological process* whereby determinate linguistic signs emerge from an inner awareness of deeper, more indeterminate forms of semantic potential. The theory of *sphoṭa* therefore already implies that the practice of 'languaging' requires a certain *intentional form of continuity*, one that the Pratyabhijñā will come to identify with the unity of *vimarśa* ('reflexive consciousness') and *prakāśa* ('manifestation').

### 3. Sambandha as a 'Śakti-of-Śaktis'

As should be clear, Bhartṛhari's three-fold progression of linguistic self-disclosure suggests a certain conception of the way *relationality* functions in an intentional/linguistic context. In particular, the ideas in the VP and the SSam prefigure the important distinction fleshed out in Utpala's SS between the relational *tokens* of conventional speech-acts (*dhvani-śabdha*) and the universal *type* (i.e., 'relationality-as-such') that characterizes a holistic *sphoṭa*'s active transformation into a particular sequence of sounds.[23] This means that the way in which relational forms appear *in* language ('horizontal relations') must be logically distinguished from the general form that characterizes the intentional process of '*languaging*' *itself* ('vertical relation'). In other words, *sambandha* comes to denote a *transcendental* principle, one whose *activity* (*śakti*) informs the 'syntactical' realization (*prakāśa*) of *sphoṭa* at an intentional level of description.[24]

Bhartṛhari comes to this theory of *sambandha-qua-śakti* in the SSam through investigating the question: 'how can 'relation' be expressed in words?' (Houben: 98) In the opening two verses of the SSam, he proclaims that there is a 'well-established' relation between words and the object of understanding (i.e., the *vācyavācakabhāva*).[25] In verse 3, he proceeds to pinpoint the genitive relation as the means by which we know that a referent (*artha*) is signified by a word (*śabda*) (e.g., as expressed in the propositions 'this is the signifier of this, and this is the signified of this').[26] Since we cannot practically deny that we have knowledge of the designation of a thing while also using and understanding the signification function (viz., semiotics) of language, Bhartṛhari suggests that some general relational capacity is presupposed in the intentional and linguistic structure of all qualified cognitions.[27]

In the next two verses (4–5), however, he identifies a perennial problem that ensues when we attempt to identify *sambandha itself* as a substantial term through an ascription of some genitive or predicative properties. In brief, we cannot exhaustively characterize 'relation' in linguistic terms because it will always be presupposed in the capacity that enables speakers to identify things through the conceptual ascription of linguistic properties. Bhartṛhari interprets this to mean that the 'extremely dependent' form of *sambandha* cannot be designated through words:

> There is no word that signifies the relation according to its specific property. Because it is extremely dependent (*atyanta-paratantra*), its form cannot be pointed out. Where this [relation] is, because some service is rendered [from one thing to another, or: from signifier to signified and vice versa], there one arrives at a property (viz. dependence) [but not at the relation itself]. It is even a capacity (i.e., something dependent) of capacities [which are themselves dependent upon the entity which possesses the capacity]; it is even a quality (i.e., something dependent) of qualities [which are themselves dependent upon the entity which possesses the quality] [so it is extremely dependent].[28]

Bhartṛhari describes relation, in its general form, as 'extremely dependent,' in the sense that it cannot *itself* be signified in words. When we cognize some limitative 'service rendered' (*upakāra*) between phenomena, we 'infer' (*anugamyate*) that a property such as 'dependence' has occurred.[29] But this inferential 'property' is always necessarily one step removed from the *śakti* of relation *in and of itself*, because the *capacity* to designate this relational property must already be in place for the designation itself to occur.[30] Thus, while a relation incontrovertibly exists between signifier and signified, no terms can designate or characterize what the specific property of this relation *is*. Houben summarizes: 'Therefore, it is never possible to isolate the relation itself from the things that are connected through the relation, and, consequently, it is not possible to attribute specific attributes or qualities to the relation itself, without transforming it into something which it is not . . . In other words, if one reifies the relation it is not a relation anymore' (174).[31]

This conception of relation as a form of *śakti* inspires Bhartṛhari's novel reinterpretation of *samavāya*, the 'inherence' relation, in verses 8–12 of the SSam. Recall that for the Vaiśeṣika relational realists, *samavāya* is an independently existent category, often de-

scribed as the 'glue' that binds together the other independent categories of existence.[32] Kaṇāda's canonical definition of *samavāya* is 'that which produces the cognition 'this is here' [or 'this subsists in this'] with respect to a cause and its effect.'[33] In this respect, it is that objective category which localizes, or restricts, universals to particulars. In an earlier verse (6), Bhartṛhari concurs that *samavāya* is often called 'relation' (*sambandha*) because it seems similar in virtue of its dependent status. However, he adds that this does not indicate the general capacity of *sambandha*, because *samavāya*'s function is restricted (*niyata*) to certain domains (i.e., primarily between whole and part). Thus, Bhartṛhari reasons that this referential restriction must occur in virtue of some higher-order 'service rendered'—i.e., an even *more general form* of *sambandha* that restricts the domains of relational subsets such as *samavāya*.

> That capacity (*śakti*), called *samavāya*, rendering service to capacities, beyond difference and identity (*bhedābheda*), being established otherwise, is assisted by *saṃbandha* (relation), which is beyond the attribute of all categories or objects (*padārtha*) and which is characterized by everything. This is the tradition from the ancients. By others it is taken for granted that *sambandha* be always made into a category or object. [But] with this, word meaning cannot be sorted out.[34]

This description of *samavāya* itself as a *śakti* completely departs from the Vaiśeṣika realists; no sources describe it as such. It is an innovation of Bhartṛhari's system (at least insofar as the traditional authority of 'the ancients' referred to above remains obscure). According to Bhartṛhari, a *śakti* is something neither identical with, nor different from, its substratum (Houben: 183–4). Thus, in identifying the *samavāya* relation with *śakti*, 'beyond difference and identity' (*bhedābheda*), Bhartṛhari not only de-reifies it from its categorical status, but also renders it referentially ambiguous or vague in terms of *bhedābheda*. This is quite in opposition to the Vaiśeṣika, whose 'radical' approach to realism invariably consisted of pinpointing the exact character and properties of a category (*padārtha*), and how precisely it differs from all others. So, the delimited relation of *samavāya* already corresponds to a capacity (*śakti*) of linguistic reality that establishes the bond between universal and particular.

And yet, despite this redefinition of *samavāya* as a form of *śakti*, it is still not a 'capacity' *general* enough for Bhartṛhari's understanding of *sambandha*. In *kārikā* 11 above, he stipulates that *sambandha* itself actually 'assists' (*anugṛhṇāti*) *samavāya*. *Sambandha* accordingly becomes a capacity of an order higher than mere inherence, and thus of all qualified cognitions; it is '*sarvalakṣaṇa*' which means, following Helarājā, that each and every object characterizes and defines *sambandha* (Houben: 184.). Houben remarks, '[t]o say that *sambandha* 'relation' is 'defined by everything' amounts . . . to saying that it is not defined, not delimitated [*sic*], not characterized by a specific, restricted domain in which it occurs. It is therefore just another way of expressing the complete absence of a specific own character of relation' (185). When we attempt to reify relation, we are left with a set of abstractions that correspond to a gradation of capacities that 'render service' to other capacities, but we never 'bottom-out' at a determinate, self-standing referent that is '*sambandha*.' The abstract status of these phenomenal capacities corresponds to the restriction of semantic domains associated with the relata in that domain: *sambandha* (*śakti-of-śaktis*) → *samavāya* (*śakti*) → other *śakti*s (Houben: ibid.). The upshot is that as the definition of *sambandha* becomes more and more *abstract* (viz., moving from tokens of 'relation' to its general form), rather ironically, it eventually becomes coextensive with the *concrete capacity* that enables meaning and reference to occur in the first place.

In this concrete capacity of 'languaging', *sambandha* is a peculiar form of intentional activity that interprets signs through the designation of other signs (i.e., 'semiosis'); yet, for that very reason, its nature cannot be represented thereby.[35] The ultra-dependent form of relation, in other words, entails that it is not an autonomously intelligible 'critter' (cf. Phillips: 23; Dunne: 44–5); viz., it cannot be isolated from its relata and remain itself, for any *attempt* to reify 'relation' as an independent term in judgment will *presuppose* what it tries to reify in the very *act of reification*. Houben concludes that we 'end up with a relation

which cannot be expressed or signified as it is, which is indeterminate and to all intents and purposes without independent character' (190). In direct contrast with the Buddhist nominalist and Vaiśeṣika naïve categorical realists, then, Bhartṛhari does not believe that 'relation,' in its most general form, can be identified with any *specifiable* relational 'property' of dependence (*pāratantrya*), or inherence (*samavāya*), etc. Rather, *saṃbandha* is intentionally disclosed as an all-pervasive and continuous '*śakti*'—the transcendental capacity of *śabda-brahman* to manifest the diversity of the world in and through the practical use of language.

### 4. The Buddhist Denial of Real Relations

Before explaining Utpala's form of relational realism—and Bhartṛhari's influence thereupon—it would help to briefly outline the Buddhist position, Utpala's *pūrvapakṣin* in the SS.[36] In the ĪPK I.2.11, Utpala summarizes the reasons for the Buddhist rejection of a realist conception of relation, alluding to Dharmakīrti's deconstructive arguments in the SP:[37] '[There is no such thing as (*saṃbandha*)] because its form, [which is based on] a dual-locus, is not unitary, and because what is [already] actualized (*siddhasya*) cannot 'require' another, and dependence, etc., do not make sense. Thus, the agent is also a conceptual construct (*kalpita*).'[38] Essentially, Dharmakīrti is arguing that *saṃbandha* is something situated in two places (*dviṣṭha*).[39] However, since, according to Dharmakīrtian nominalism, all real momentary entities are 'self-contained' and 'reside in themselves,' the 'distributed,', or 'continuous' (*anvaya*), nature of the *saṃbandha* necessarily undermines its status as an ultimately real entity.[40] For once a particular moment comes to *exist* in the causal procession, instantiating a self-contained essential nature (*svabhāva*), it ipso facto requires no *dependence* merely to do so. We should mention that the same argument applies, mutatis mutandis, to notions of continuous *action*, which unify a diversity of moments into a single continuous form.[41] Thus, Dharmakīrti himself views the definitions of relation and action as tightly intertwined, concluding on this basis that the relative notion of an agential self—i.e., one upon whom the realization of distinct actions *depends*—is ultimately also a conceptual construction.[42]

In the SS, Utpala expands upon the Buddhist reductionist position he introduces in the ĪPK, such that the simultaneously unified and diverse form of 'relation' is simply a flagrant violation of the law of the excluded middle: 'Furthermore, a unity of many that is coextensive with multiplicity does not make sense, because [it is] incompatible with the nature of existence and non-existence. There would arise (*udbhavet*) either unity from a manifold material cause (*anekasmād upādānād ekam*), or a manifold from a unity—insofar as they are only mutually unrelated, they would resolve simply in themselves.'[43] The Buddhist therefore sees no difference between the assertions 'X is both unified and diverse' and 'X both exists and does not exist'. If X exists as a simple (*ekātva*) particular, it definitionally does not participate in the identity of any other real existent, Y.[44] Any ascription of these predicates to a phenomenon must therefore be *sequential* properties of distinct moments, not simultaneous attributes of a substantial (*dravyatas*) existent.

In effect, then, the Buddhist demonstrates that the axioms of nominalism and its dichotomous ontology—i.e., between ultimately real *svalakṣaṇa* and merely conventional *sāmānyalakṣaṇa*—automatically pigeonhole *saṃbandha* into the same category as *universals*, for the distinction between non-relational particulars and conceptual elaboration associated with the determinate language of discursive thought necessarily entails that 'relations' only appear in the *latter* moment. Thus no genuine *continuity* between the two moments can obtain, for all relations are attributed post hoc to the disconnected sequence.[45] As Utpala's Buddhist concludes, 'Hence, only at the time of conception (*pratītikāla*) do we accept a relationship as though universal.'[46] Since both are 'distributed/continuous' (*anvaya*) entities (Dunne: 80), they are both not ultimately real—viz., *they only appear in the moment of determinate conceptual elaboration* (*kalpanā*), not the initial indeterminate, non-relative moment of perceptual cognition (*pratyakṣa*).[47] The conceptual dichotomy between the 'unity' and 'diversity' *as predicates of a substantial existent* therefore cannot characterize

the real momentary entity, for each numerical property can only appear sequentially, characterizing distinct phenomena in the causal procession.

I argue below that Utpala replaces the implicit Buddhist identification of the categories of *sambandha* and *sāmānyalakṣaṇa* with Bhartṛhari's identification of *sambandha* and *śakti*. That is to say, the general form of relation, or unity-in-diversity, is not a predicative *property* of a *substantial existent*, one that we can characterize or describe in terms of *other determinate concepts*. Rather, it is coextensive with a transcendental form of *action* whose intrinsic continuity ultimately makes the intentional form of synthetic judgment—and thus *practical reference to universal entities*—possible in the first place. In terms of the relational axes mentioned earlier, this redescription of *sambandha* represents a movement from the Buddhist's 'horizontal' assessment of 'relations' in terms of various conventional tokens (i.e., those experienced solely in the moment of *vikalpa*) to an analysis of the general type (i.e., 'relationality-as-such') which manifests 'vertically' in the continuous action of *vimarśa*, or recognitive judgment.

## 5. Utpaladeva's Pragmatic Elaboration of Sambandha-Śakti

Although Bhartṛhari is not explicitly mentioned, we can begin to appreciate his influence on Utpaladeva with reference to the general methodology of the SS. That is, Utpaladeva does not marshal arguments for any subset of relations (e.g., *pāratantrya*, *kāryakāraṇabhāva* and *samavāya*), but instead seeks to articulate, and prove to be real (*satya*), the existence of a *general* form *of sambandha*. According to Utapala, the common feature of every 'relation' (e.g., 'mixture' (*saṃsarga*), 'contact' (*saṃparka*), 'fusion' (*saṃśleṣa*), 'dependence' (*pāratantrya*) and 'requirement' (*apekṣā*)[48]) is nothing other than unity-in-diversity (*bhedābheda*).[49] Like Bhartṛhari, then, Utpala abstracts from all tokens of *sambandha* to get a handle on its general logical form, and then infers that it is irreducibly characterized as being neither wholly identical with, nor wholly distinct from, its respective relata.

After the Buddhist lambasts this idea as theoretically incoherent (on the basis of the nominalist modes of reasoning just outlined), Utpala cites none other than Dharmakīrti himself to show that the Buddhists already take for granted the existence of relational cognitions that unify a diversity of relata: 'First of all, this clear apprehension (*pratīti*) is acknowledged even by you [Buddhists] for the sake of designating a relation that pertains to the form of a unity of many, such that it is said: 'Thus entities themselves are disjunct; conceptual construction conjoins them' [SP 5cd]. That its form is a conceptual construction, though (*ca*), is not a problem. Even the distinction (*vikalpa*) 'this is a pot,' 'this is a cloth' is just a conceptual construction (*kalpana*).'[50] This passage articulates Utpala's movement of the conversation into a 'transcendental' register, in the sense that the real issue at hand is not Dharmakīrti's justified skepticism towards the reality of any intentional *content*, but an incoherent skepticism toward the *synthetic* form of *cognition itself*—i.e., the 'capacity' that lends conceptual determinacy to this content. For Utpala, in other words, the 'problem' with unity-in-diversity is not that relational cognitions are *conceptual constructions*, the 'problem' is rather in thinking this means they are not *real* (*satya*).[51] Dharmakīrti purportedly wants to consider relations to be a form of conceptual 'error' (*bhrānti*) or 'defect' (*doṣa*)—that is, a *superimposition* upon a reality that is *constitutively otherwise* (i.e., momentary, non-conceptual, impersonal and non-relational). But Dharmakīrti takes for granted that designating perceptual objects *as particulars* already instantiates a synthetic form of cognition—and, most importantly, he does not consider *this form of cognition an error*.

Of course, Utpala's Buddhist acknowledges that he does not regard the synthetic form of these *perceptual* judgments of particular objects as *purely* conceptual, due to the causal efficacy of the ultimately real percept: 'In this case, there is also a perceptual appearance such that (*tathā*) there is precisely the form of [determinate, individual] things like pots and cloths, etc.; hence it [i.e., the appearance] is not a conceptual construct.'[52] This statement reflects the prototypical Buddhist position of the causally privileged nature of perceptual cognitions—even if the relations that characterize the designation are not real, the perceptual 'data' still instantiate the self-standing nature of ultimately real objects.

Insofar as the content of these perceptual judgments is *a particular object*, the cognition is not *just* a synthetic conceptual construction.

But Utpala argues on constitutively *practical* grounds that Dharmakīrti is not entitled to infer from the conceptual relations that distinguish objective *content* that the synthetic form of cognition *itself* is consequently *unreal*. That is to say, there is no *practical* difference between the *general* conceptual form of relations that distinguish particular objects from each other (i.e., an individual 'pot' and 'cloth') and the general form that happens to *refer* to a conceptual relation (i.e., 'the king's servant' and 'rice in a pot'[53]). In Utpala's view, this imagined dichotomy is based on a category mistake that conflates the 'external' tokens of relation that *define* the contentful appearance of perceptual objects with the 'internal' relations that practically *constitute* the synthetic form of judgment.[54] After all, Utpala insists, even the practical recognition of an empirical *error* presupposes a synthetic form of judgment, insofar as it requires a cognition of *similarity* between two *different* appearances—i.e., the universal idea *previously overturned* and the percept currently *doing the overturning*[55]: 'For it is not because of its being closely connected with the appearance of an object that it [i.e., the cognition of relation] is [difficult to be uprooted] by a defeater; rather, it is just because of the real existence of *similarity* among appearances of objects. Even in the case of the illusion of silver, there actually is the real presence of similarity with mother-of-pearl: For all misapprehensions (*bhrānti*) have *similarity* as their content—and some of those (*cānyā*), by excluding similarity, make sense only as being closely connected to the appearance of an object.'[56]

For Utpala, the Buddhist misconstrues the practical significance of his own observation—i.e., that some perceptual 'errors' (such as relation and partite extension) are difficult to practically uproot because they are 'closely attached' to the appearance of the object, for this 'close attachment' is not an instance of conceptual *error*, but rather *just what it practically means* for a particular object to appear as it does in perceptual judgment. In other words, the peculiar phenomenal feature of the appearance of particulars is not that it picks out some 'causally privileged' percept that abides wholly *unrelated* to this synthetic act of conceptualization. Rather, it is that the apprehension of an appearance *as* an external particular *simply consists in* the conceptual *exclusion* of everything that appears *as similar* thereto under some practical description.[57] For Utpala, the phenomenal particularity of any given object is continually 'pinpointed' in consciousness through the conceptual process of *apoha*, which excludes appearances according to their relevance for the determinations necessary to make sense of its own practical activity.[58] On pragmatic grounds, therefore, Dharmakīrti cannot appeal to a judgment of some non-relational particular to discount the validity of the perception of relations such as 'the king's man' and 'rice in a pot,' even though they are conceptually determined. For, just like the appearances of the particular objects, these relational cognitions are never practically overturned, even though they exhibit a form of unity-in-diversity.[59]

According to Utpaladeva's system of absolute idealism, then, this general form of relation, unity-in-diversity, can only manifest due to the reflexive nature of self-consciousness that spontaneously (*svātantrya*) re-cognizes a manifold of perceptual appearances in conceptual terms.[60] This is the true significance of relationality for Utpala, as he states in the ĪPK II.2.4: 'Things that are self-contained and manifested separately possess a unity in the sense of mutual connection within the unitary knowing subject. This is the basis of the ideas of relation.'[61] We can say, therefore, that Utpala regards *all tokens* of *sambandha*—from grammatical to causal—as necessarily instantiating the general unity-in-diversity of reflexive *judgment*. This proves the Buddhist cannot appeal to the apparent distinctions between particular objects to establish the causally privileged (read: ultimately real and non-relational) status of perceptual cognitions, because the referents of both grammatical and causal relata are determined through the same synthetic form of cognition.

## 6. Judgment and the Practical Determination of Relational Tokens

As we have seen, Utpala argues that all relations exhibit a general form of *bhedābheda*. At some point in the SS, he entertains a possible Buddhist objection that this theory merely recapitulates the theoretical problems of the Vaiśeṣika category of *samavāya*, for this 'unity-in-diversity' supposedly means a single, general form relates *everything*, which either entails the familiar regress of Dharmakīrti or the inability to account for the diversity of relational tokens. But Utpala retorts that 'this would be the case if a relation were to consist *exclusively* of unity; [but] a relation is a unity *with parts* to the extent (*yāvatā*) that the parts are distinct. And therefore, the unity that is 'tinged' (*uparakta*) by "the king's servant" is a *different* relation from the one implied (*upalakṣaṇa*) in the 'color' (*uparāga*) of "father and son."'[62] In other words, Utpala avoids the regress of relational reification by upholding Bhartṛhari's conception of relational tokens as dependent upon the nature of the appearance of relata. No two particular relations possess *precisely* the same 'color,' because the particular form of the relata determines the nature of the relation.

But, for Utpala, the 'relation' between the king and servant depends upon the conventional circumstances in which they are investigated, for even the perceptual determination of the *relata themselves* ultimately depends upon the *practical* capacity of reflexive judgment (*vimarśa*) to take any given appearance *as* either relational *or* non-relational.[63] In other words, the relational *content* of all qualified cognitions—even relative *absence* and *otherness*—spontaneously conform to the practical interests and intent of the *agent* in virtue of the recognitive capacity of *vimarśa.* To prove his point, Utpala appeals to the rules of grammar wherein one can interpret distinct referents as a single compound word (*dvandva*).[64] For instance, in the phrase '*rājñaḥ puruṣa*' ('the king's servant'), the king is a distinct word with its own genitive case ending, and thus a dyadic relation obtains between *distinct* terms. However, in the case of '*rājapuruṣa*' (i.e., king-servant), the king 'appears (*prathate*) as having completely (*ekāntena*) assumed (*āpanna*) the identity of the qualified, its own form being altogether relinquished. Thus, in this case, there is no sense in expressing a relation.'[65] Likewise, in the case of simple predication, the appearance 'blue-lotus' could either be relational (i.e., 'the lotus is blue') or non-relational (e.g., 'the blue-lotus is not white'), depending on whether we wish to take the appearance of 'blueness' *as distinct* from the lotus.

Indeed, Utpala maintains that for *any* qualified form of relational cognition (he includes those of absence, difference and propositional *kārakas*[66]), its individual terms, and the grammatical relations that define their intentional appearance, always already disclose the existence of the capacity for them to be interpreted in either way (i.e., as a relatum or a term). Theoretically, then, the givenness of any appearance(s) necessarily instantiate an infinite capacity for relational determination; viz., any single appearance contains within itself an unbounded potential for, say, predication or relative absence. Which of these conceptual possibilities comes into clear and distinct awareness is precisely that which judgment renders intelligible for the sake of some practical orientation. There is thus an incipient or nascent form of infinite interpretability even in the act of perceiving a single, unitary object.[67] In the Pratyabhijñā system, this only makes sense because the reflexive, self-luminous consciousness of the agent (i.e., the *ātman*) spontaneously enacts relational tokens that render intelligible the diverse, indeterminate factors that enter into its own practical activity.[68]

In his effort to rebuff the Buddhist nominalist, Utpala extends this reasoning to the causal relation, for even the simultaneous appearance of a *distinct sequence* of 'relata'—i.e., in the 'metaphorical' (*gauṇavṛtti*) sense of 'assister' and 'assisted'[69]—must indicate an act of conceptual synthesis, a certain form of unity-in-diversity.[70] This form of relational judgment presents two temporally distinct moments *as* simultaneous for the sake of those practical endeavors, the conceptual synthesis of distinct appearances upon which all conventional life ultimately depends. Pertinently, to buttress his point, he invokes an argument initially deployed by Bhartṛhari[71]: For one running along a path (or, more generally, engaged in

any task where a manifold of perceptual data go consciously unrecognized) perceptual features of the surrounding environment (e.g., the grass on the side of the road) remain phenomenally unestablished. However, one can later recall having experienced these things, but only insofar as they appear in a conceptual form expressible in terms of judgment.[72]

For Utpala, the fact that a subsequent mnemonic cognition can 'unfold' the many indeterminate perceptions which were 'enfolded' into the determinate conceptual judgment at the time of activity demonstrates that 'consciousness does not passively record sensory data but appropriates them through some kind of silent expression that can later be elaborated on when (s)he thinks back on what (s)he has done and explains it. This means that discursive thought is nothing but the development of a subtle, condensed expression already present in any act of perception' (Ratié 2021, p. 96).[73] According to Utpaladeva, then, there is no hard and fast distinction between the indeterminate moment of initial perception and its subsequent conceptual determination in the stream of consciousness; the 'clear and distinct' quality of perceptual judgments already presupposes the transcendental operation of a continuous form of *sambandha-śakti* that *gives* relations in precisely the way they are self-consciously *taken.* Utpala's invocation of Bhartṛhari therefore suggests that even though he does not adopt the latter's theory of *sphoṭa* wholesale, he concurs that an implicit continuity of discursive activity defines a progressive conceptual determination of perceptual content.

Summarizing the influence of Bhartṛhari on the Pratyabhijñā, Torella comments, 'in order to undermine the discontinuous universe of the Buddhists, [Utpaladeva] decides to avail himself precisely of the [doctrine of] the language-imbued nature of knowledge, which is meant to demolish its main foundation stone, the unsurpassable gulf between the moment of sensation and that of conceptual elaboration, representing, as it were, the very archetype of the Buddhist segmented reality' (Torella 2008, pp. 350–51). We can now appreciate not only the significance of Bhartṛhari in this transcendental reformulation of relational realism, but also why the topic of *sambandha* was so central to Utpaladeva that he decided to revisit the subject on its own in the *Siddhitratyī*; for the problem of proving the reality of a dynamic, unified self against the Buddhist nominalist coextends with establishing a *continuity* between the initial moment of bare perception and its subsequent conceptualization—and only an objective, general form of relationality could provide a sufficient condition to establish such a feature of reflexive consciousness.

## 7. Conclusions

I have argued that Bhartṛhari's 'emanationist' metaphysics of *sphoṭa* (i.e., the capacity of divine 'speech' (*paravāk*) to realize itself in determinate, delimited expressions) demanded a novel reinterpretation of *sambandha* as a transcendental form of *śakti*—a '*śakti*-of-*śakti*s'. According to Bhartṛhari, while we might designate a particular token of 'relation' through a universal term (e.g., *pāratantrya*, *samavāya*), all these forms derive from the relational *capacity* of *paravāc* that makes the very *act* of linguistic reference—and thus all qualified knowledge—possible in the first place. We have seen that the Pratyabhijñā non-dualists, in the service of establishing their own absolute idealism, appropriate and develop this identification of *sambandha* and *śakti* into a series of practical arguments against the Buddhist relational eliminativist: in the SS, Utpala argues that the apparent unity-in-diversity of relations cannot be a form of conceptual 'error' (*bhrānti*), like universals, because the practical appearance of any conceptual 'errors' *itself* depends upon the synthetic form of practical judgment. He then proceeds to claim, on the basis of grammatical and phenomenological considerations, that this general synthetic form of judgment determines the appearance of any given referent *as* either relational or a separate term in yet another predicative relation. In other word, no 'horizontal' linguistic designation (i.e., the distinct relational tokens of particular judgments) can ever represent the 'vertical' capacity to *interpret* intentional content in *those very terms* (i.e., as either relational or non-relational).

In the end, the Buddhist nominalist cannot *deny* the unity-in-diversity of the general form of relation on purely *theoretical* grounds without thereupon casting asunder the

*practical* capacity to determine a manifold of perceptual appearances through the act of conceptual designation. And, since the Buddhist presumably takes this capacity for granted even in the perceptual judgment of *particular objects*, he cannot invoke our awareness of these very appearances to justify the constitutively non-relational character of the ultimately 'real.' In sum, then, Bhartṛhari's reformed conception of *sambandha* contributes greatly to the Pratyabhijñā objective idealist project, which seeks to leverage transcendental arguments to move beyond the skeptical representationalism and epistemic idealism that the dichotomous ontology of Buddhist nominalism invariably engenders.[74]

**Funding:** This research received no external funding.

**Institutional Review Board Statement:** Not applicable.

**Informed Consent Statement:** Not applicable.

**Data Availability Statement:** Not applicable.

**Conflicts of Interest:** The author declares no conflict of interest.

## Appendix A

### Sanskrit Texts

[ĪPV] Abhinavagupta. *Īśvarapratyabhijñāvimarśinī.* Edited by K. A. Subramania Iyer and K. C. Pandey. Motilal Banarsidass, 1986.

[ĪPK] Utpaladeva. *Īśvarapratyabhijñākārikā* and *Vṛtti.* See (Torella [1994] 2013).

[ĪPVv] Utpaladeva. *Īśvarapratyabhijñavivṛti.* See (Ratié 2021).

[VP] Bhartṛhari. *Vākyapadīya.* See (Iyer 1963, 1966) and (Rau 1988).

[VS] Kaṇāda. *Vaiśeṣika Sūtra.* See (Chakrabarty 2003) and (Sinha 1910)

[ŚD] Somānanda. *Śivadṛṣṭi.* See (Nemec 2011).

[SSam] Bhartṛhari. *Sambandhasamuddeśa.* See (Houben 1995a).

[SP] Dharmakīrti. *Sambandhaparīkṣā.* See (Steinkellner 2021) and (Jha 1990).

[SPV] Devendrabuddhi. *Sambandhaparīkṣāvṛtti.* See (Steinkellner 2021) and (Frauwallner 1934).

[PV] Dharmakīrti. *Pramāṇavārttika.* Y. Miyasaka, *Pramāṇavārttikakārikā* (Sanskrit and Tibetan). *Acta Indologica* 2 (1971–1972), 1–206.

[MBhD] Bhartṛhari. *Mahābhāṣyadīpikā.* See (Abhyankar 1983).

## Notes

[1] For work that touches upon Bhartṛhari's influence on Kashmir Śaivism, cf., Iyer (1969); Dwivedi (1991); Torella ([1994] 2013, pp. xxiv–xxv; 2008); Rastogi (2009); Ratié (2011a, 2018); Vergiani (2016); and Ferrante (2017, 2020a, 2020b).

[2] On these topics, see Torella (2008), Rastogi (2009) and Ferrante's (2017, 2020a, 2020b, 2020c) recent studies on Bhartṛhari and Pratyabhijñā.

[3] Aside from a chapter in Allport's (1982) Oxford thesis on Utpaladeva, see MacCracken's (2017, 2023) recent publications on the SS, which have contributed to this endeavor (none of these sources, though, focus exclusively on Bhartṛhari's influence on the Pratyabhijñā theory of *sambandha*).

[4] I should add that I have not yet seen MacCracken's (2021) dissertation where he translates the SS in full; we were working on translations of the text at about the same time. My own dissertation (forthcoming in 2024) translates, analyzes and compares Dharmakīrti's SP and Utpaladeva's SS.

[5] While the reality of *sambandha* indirectly impacts many of the arguments in the ĪPK, they are only an explicit object of discussion in section II.2.1–II.2.7.

[6] My analysis of the SSam is heavily indebted to Houben's comprehensive (1995) study of the text.

[7] For Pratyabhijñā as transcendental philosophy see Lawrence (1999, 2019). Much of what I will say here vis-à-vis the appropriation of Bhartṛhari's conception of *sambandha* can also contribute to comparative conversations around the semiotics of Peirce, in which the mediating function of 'Threeness' functions much like the Pratyabhijñā conception of recognitive *śakti*. (On Peircean semiotics and Pratyabhijñā, see Lawrence (2018a, 2018b) and MacCracken (2023)).

[8] More specifically, the Pratyabhijñā thinkers add a fourth, more supreme stage—*parāvāk*—to Bhartṛhari's three-fold scheme I will outline later in the paper. While the details are not important here, the addition of this fourth level, associated with the ultimate, undifferentiated essence of Śiva himself (cf. ĪPK 1.5.13), symbolizes how Utpala intended to both defend and subsume

Bhartṛhari's linguistic metaphysics into his own system. See Torella (2001, pp. 857–59) and Prueitt (2017, pp 80–83) for further discussion on these stages of speech in Pratyabhijñā, and Ferrante (2020c) and Torella (2001) on aspects of denotation (*abhidhā*) in the Pratyabhijñā system.

9    This is famously expressed in the opening four verses of the VP: *anādinidhanaṃ brahma śabdatattvaṃ yad akṣaram/vivartate 'rthabhāvena prakriyā jagato yataḥ//ekam eva yad āmnātaṃ bhinnaśaktivyapāśrayāt/apṛthaktve 'pi śaktibhyaḥ pṛthaktveneva vartate// adhyāhitakalāṃ yasya kālaśaktim upāśritāḥ/janmādayo vikārāḥ ṣaḍ bhāvabhedasya yonayaḥ//ekasya sarvabījasya yasya ceyam anekadhā/ bhoktṛbhoktavyarūpeṇa bhogarūpeṇa ca sthitiḥ* (cf. translations in Reich (2021, p. 49) and Bronkhorst (1992)). I should note that Houben (1995a, pp. 16–8l; 1995b) argues that Bhartṛhari does not have any particularly strong religious or polemical commitments in the VP and employs a form of 'perspectivalism' whose primary aim is to systematize and describe the views of others. According to him, we should not necessarily read the opening theological *kārikās* as a personal statement of his own metaphysical position. Cardona (1999, pp. 92–93), on the other hand, contends that this interpretation is unwarranted and does a disservice to the grammatical tradition with which Bhartṛhari identifies (cf. Todeschini (2010, fn. 14)).

10   Bhartṛhari distills his metaphysics of language in VP 1.131: *na so 'sti pratyayo loke yaḥ śabdānugamād ṛte/anuviddham iva jñānaṃ sarvam śabdena bhāsate* (Iyer (1966)).

11   The sacredness of sound and speech, of course, trace back to the *Ṛg-Veda*. Two hymns, 10.125 and 10.71, are dedicated to *vāk*. Pertinently, while the former verse personifies Vāk as a goddess, the latter discusses three stages in the development of language: (1) inarticulate speech (e.g., the sounds of animals and insects), (2) primitive articulate speech (i.e., nominal designation), and (3) discursive, or 'proper' language that represents the 'refined' (*saṃskṛta*) form of the Vedic sages and poets (Beck 1993, p. 37).

12   VP I.94: *avikārasya śabdasya nimittair vikṛto dhvaniḥ/upalabdhau nimittatvam upayāti prakāśavat.*

13   Bhartṛhari describes this progressive unfoldment of *sphoṭa* (VP 1.83–1.86 and *svavṛtti*) as a maturation of conceptual 'seeds' in the mind consisting of 'memories, effects, and potentials' (*saṃskāra-bhāvanā-bījāni*). The semantic determinacy of these inchoate realities intentionally resolves through a series of sounds and ultimately terminates with the utterance of the final sound that discloses the own-form of a word. (See, in particular, VP 1.86 and its *svavṛtti*: *nādair āhitabījāyām antyena dhvaninā saha/āvṛttaparipākāyāṃ buddhau śabdo 'vadhāryate. nādaiḥ śabdātmānamavadyotayadbhir yathottarotkarṣeṇādhiyante vyaktaparicchedānuguṇa saṃskāra-bhāvanā- bījāni/tataścāntyo dhvaniviśeṣaḥ pariccheda-saṃskāra-bhāvanā-bīja-vṛtti-lābhaprāptayo- gyatāpripākāyāṃ buddhāvupagraheṇa śabdas- varūpākāraṃ saṃniveśayati*).

14   Quoted in Monier-Williams et al. (1970, p. 1270). Patañjali also characterized *sphoṭa* as possessing unity, indivisibly and eternality (Chakravarti 1930, p. 89).

15   See, e.g., Coward (71) and Matilal (85).

16   Bhartṛhari makes this clear in the VP when he discusses the important idea of *pratibhā* (Tola and Dragonetti (1990, p. 97); Desnitskaya (2016, p. 330). This term is variously rendered as 'intuition' (Coward and Raja 1990 (144)) or 'flash of understanding' (Akamatsu 1992, pp. 37–40)), but, in any case, represents the apprehension of the holistic nature of the *vākya-sphoṭa* over and above its constitutive *pada-sphoṭa*: *vicchedagrahaṇe 'rthānāṃ pratibhānyaiva jāyate vākyārtha iti tām āhuḥ padārthair upapāditām//idaṃ tad iti sānyeṣām anākyeyā kathaṃ cana pratyātmavṛtti siddhā sā kartrāpi na nirūpyate//upaśleṣam ivārthānāṃ sā karoty avicāritā sārvarūpyam ivāpannā viṣayatvena vartate//sākśāc chabdena janitāṃ bhāvanānugamena vā iti kartavyatāyāṃ tāṃ na kaś cid ativartate//pramāṇatvena tāṃ lokaḥ sarvaḥ samanugacchati samārambhāḥ pratāyante tiraścām api tadvaśāt* (VP 2.143–7; cf. translation in Akamatsu (1992, ibid.)). In this sense, it has been noted that Bhartṛhari advocates a theory of semantic holism similar to both Frege (1892) and Quine (1951), where the semantic content of propositions cannot be derived bottom-up from their constitutive elements (Chakrabarti (1989) presents an analysis of sentence-holism in the Indian context). It is noteworthy that Dharmakīrti rejects this idea when he reduces sentence meaning to its constitutive words (in, e.g., PV I.127.2–5), which aligns with the nominalist tendency to view sentence meaning as a conceptual entity whose distributed (*anvaya*) character entails that, ipso facto, it cannot be ultimately real (Dunne: 80–83). For a highly relevant description of *pratibhā* as a form of *practical* knowledge, see recent articles from David (2021) and Das (2022) (I must thank an anonymous reviewer for bringing my attention to these works).

17   *dvāv upādānaśabdeṣu śabdau śabdavido viduḥ/eko nimittaṃ śabdānām aparo 'rthe prayujyate* (VP 1.44).

18   Cf. MBhD (49): *karaṇasannipātāt*. See Seneviratne (125–139) for an extended discussion of the *dhvanis* and, in particular, Bhartṛhari's binary division of *dhvanis*: '[T]he "*prākṛta*"(original, therefore, "primary") type of *dhvani* is what reveals *sphoṭa*, while the "*vaikṛta*" (evolved from another, therefore, "secondary") type of *dhvani* maintains the continuity of the already revealed *sphoṭa*' (Seneviratne, p. 126).

19   Cf. Houben (1995a, preface). *The Saṃbandha-Samuddeśa (Chapter on Relation) and Bhartṛhari's Philosophy of Language.*

20   This intermediate stage is technically characterized by internality (*antaḥsaṃniveśa*), mind-dependence (*buddhimātropādāna*) and sequentiality (*parigṛhītakrama*). Ferrante (2020a, pp. 149–50) suggests that these three qualities can be tentatively compared to a sort of Fodorian 'mentalese' (1975)—viz., a 'Language-of-Thought' (LoT) theory that posits a type of higher-order formal language, distinct from any determinate public language, that defines the universal syntactic structures of symbolic representation. Obviously, Fodor would likely have no truck with Bhartṛhari's third level of *paśyanti vāc* that embraces a monistic vision of linguistic emanation. But one of the challenges that Bhartṛhari and the holism of *sphoṭa* can debatably present LoT theorists is to explain the experienced *continuity* between the higher-order LoT and the first-order diversity of public languages—and,

metaphysically speaking, the continuity of the 'intentional' dimension of the LoT with the even lower-level domains of regular order present throughout the empirical world.

21   See *vṛtti* on VP 1.159: *paraiḥ saṃvedyaṃ yasyāḥ śrotraviṣayatvena pratiniyataṃ śrutirūpaṃ sā vaikharī. śliṣṭā vyaktavarṇasamuccāraṇā prasiddhasādhubhāvā bhraṣṭasaṃskārā ca. tathā yā 'kṣe yā dundubhau yā veṇau yā viṇāyām ity aparimāṇabhedā. madhyamā tv antaḥsaṃniveś- inī parigṛhītakrameva buddhimātropādānā. sā tu sūkṣmaprāṇavṛttyanugatā kramasaṃhārabhāve 'pi vyaktaprāṇaparigrahaiva keṣāñcit. pratisaṃhṛtakramā saty apy abhede samāviṣṭakramaśaktiḥ paśyantī. sā calācalā pratilabdhasamādhānā cāvṛtā ca viśuddhā ca, sanniviṣṭajñeyāk- ārā pratilīnākārā nirākārā ca, paricchinnārthapratyavabhāsā saṃsṛṣṭārthapratyavabhāsā praśāntasarvārthapratyavabhāsā cety aparimāṇabhedā* (cf. Ferrante (2020a, p. 149, f.5)).

22   The circular nature of encoding and decoding *sphoṭa* is evocative of Bohm's process metaphysics of implicate wholeness (1980). In these terms, we might say that a semantic whole internally 'unfolds' into a diversity of particular *dhvani*, which, in turn, 'enfold' this implicate whole in the external act of communication. The potential comparison here might be theoretically fruitful for a metaphysics of wholeness, particularly insofar as Bohm (1980) bases his theory primarily on modern physics, while Bhartṛhari, of course, focuses on the intentional, or phenomenological, dimensions of language use.

23   This is among the positions he cites, but does not necessarily overtly endorse, in the VP: *anekavyaktyabhivyaṅgyā jātiḥ sphoṭa iti smṛtā/kaiś cit vyaktaya evāsya dhvanitvena prakalpitāḥ* (VP 1.93; cf. Bronkhorst (1992, p. 10), who lists this verse as I.96). Even though he does not explicitly commit to this position, I would view this as fairly consistent with his own relational project, as will become clear.

24   These two relational axes map onto Utpala's disjunction between non-sequential and sequential forms of action (see, on this point, Ratié's recent study on the ĪPKVv II.1, Ratié (2021, pp. 115–21)). In future publications, I plan to explore the way this Pratyabhijñā model can contribute to Kimhi's (2018) and Rödl's (2018) recent analytical attempts to recuperate absolute idealism. Consider Kimhi's claim that 'judgment belongs to a certain context of activity: the activity whose unity is the same as the consciousness of its unity, or self-consciousness' (Kimhi: 52). Due to the intrinsic self-consciousness of the activity of judgment, Kimhi insists that what is expressed by 'something of the form '*S* thinks that *p*' should not be logically assimilated to that which is expressed by sentences such as '*S* is doing *φ*' or '*S* is *φ*-ing' . . . . The 'thinks' in '*S* thinks' is an activity in what is logically a fundamentally different sense of 'activity' from any expressed by verbs in predicative propositions of the form '*S* is *φ*-ing.' . . . We misunderstand this uniqueness if we construe it in terms of the exceptional nature of either the substance or attributes involved in a nexus of predication' (ibid., 15–16). Depending upon how you interpret Kant, the Pratyabhijñā may be said to diverge in their conviction that nothing outside the determinative activity of self-consciousness could appear as anything within it *unless its nature is ontologically continuous with this activity*. In other words, a reified interpretation of the noumenal is discredited merely by virtue of the intrinsically self-illuminating structure of self-conscious cognition, whose sentience consists in the fact that it cannot illuminate anything fundamentally other than itself (on this point, see Arnold (2008) and Ratié (2011b, 2014)). This is also, of course, precisely where Hegel disembarks from Kant's dichotomous system of thought into a triadic metaphysics.

25   *jñānaṃ prayoktur bāhyo 'rthaḥ svarūpaṃ ca pratīyate/śabdair uccaritais teṣāṃ saṃbandhaḥ samavasthitaḥ//pratipattur bhavaty arthe jñāne vā saṃśayaḥ kvacit/svarūpeṣūpalabhyeṣu vyabhicāro na vidyate* (VP III.3.1–2). (See Houben (1995a, pp. 149–53) and Biardeau (1964, p. 423) for expositions of these two verses, and, in particular, how the three-fold relational factors mentioned in the first two verses collapse into a dyadic *śabda-artha* relation in the subsequent verses).

26   *asyāyaṃ vācako vācya iti ṣaṣṭhyā pratīyate/yogaḥ śabdārthayos tattvam apy ato vyapadiśyate* (VP III.3.1.3).

27   Houben notes that Bhartṛhari's method of 'proof' in this section is like those given in the *Vaiśeṣikasūtra*, where some very commonplace understanding serves to substantiate their categorical analysis of cognition (172, f. 280).

28   *nābhidhānaṃ svadharmeṇa saṃbandhasyāsti vācakam/atyantaparatantratvād rūpaṃ nāsyāpadiśyate//upakārāt sa yatrāsti dharmas tatrānuga- myate/śaktīnām api sā śaktir guṇānām apy asau guṇaḥ* (VP III.3.4–5; translation in Houben: 170).

29   Note that this picture roughly corresponds to Dharmakīrti's inferential explanation of the causal 'relation' in the SP (v. 13): 'Upon observing one [thing]—i.e., when something [that was] unseen is seen, and is not seen when that [other] is not seen—a person infers 'effect' even without the explanation of another'. *paśyann ekam adṛṣṭasya darśane tadadarśane/apaśyan kāryam anveti vināpy ākhyātṛbhir janaḥ*.

30   Whether Bhartṛhari considers *vācyavācakabhāva*s eternal or created is unclear (cf. VP 1.28: *nityatve kṛtakatve vā teṣām ādir na vidyate/prāṇinām iva sā caiṣā vyavasthā nityatocyate*). In any case, he states that the relation is 'permanent' (*nitya*) insofar as its 'beginning cannot be found' (*ādir na vidyate*). To the extent that the relation between universals is eternal in the same way that *universals* are, one can advance the potential connection between *sphoṭa* and *logos* (Śāstrī 1959, pp. 102–3; Beck 1993, pp. 14–15). Consider Gerson's interpretation of Plotinus's statement, 'That the Intelligibles are not Outside the Intellect and on the Good' in the *Enneads*: '[Plotinus] seems to want to argue not only that eternal Forms exist, but that these are somehow connected eternally. This is so presumably because it is owing precisely to such eternal connections that instances of Forms are necessarily connected. Thus, if x is f entails that x is g, this is because of the necessary connectedness of F-ness and G-ness. And here we must add that the eternal connection is ontologically on a par with the eternity of each Form in the connection, that is, the condition for the possibility of x being f is no more eternal than the condition for the possibility that if x is f, x must be g. At this point, Plotinus seems to be arguing that the eternal "link" between eternal, immaterial entities must be one thing which is capable of simultaneously being identified with both F-ness and G-ness so that the partial identity of these is grounded in reality . . . This one thing is what Intellect is supposed to be. Intellect must be eternal because any judgment made by an individual mind

depends for its truth on eternal reality, including eternal interconnectedness of Forms, hence eternal Intellect which grounds the interconnectedness' (Gerson 1994, pp. 48–49).

31  This observation relates to arguments put forward by Dharmakīrti in the SP and F. H. Bradley (1893) against the reality of relation. Specifically, they both observed that when we try and make any token of 'relation' into a substantive *term* with a capacity for genitive predication, we wind up with a regress, for we will always require a further relation to relate this independent relation to its own properties, ad infinitum: *dvayor ekābhisambandhāt sambandho yadi taddvayoḥ/kaḥ sambandho 'navasthā ca na sambandhamatis tathā* (SP v. 4).

32  See, e.g., *Classical Indian Metaphysics: Refutations of Realism and the Emergence of "New Logic".* (Phillips 1995), *The Lost Age of Reason: Philosophy in Early Modern India, 1450–1700.* (Ganeri 2011) *and Ascription of Linguistic Properties and Varieties of Content: Two Studies on Problems of Self-Reference.* (Oetke 2012).

33  *ihedam iti yataḥ kāryakāraṇayoḥ sa samavāyaḥ* (VS 7.2.26).

34  *tāṃ śaktiṃ samavāyākhyāṃ śaktīnām upakāriṇīm/bhedābhedāv atikrāntām anyathaiva vyavasthitām//dharmaṃ sarvapadārthānām atītaḥ sarvalakṣaṇaḥ/anugṛhṇāti saṃbandha iti pūrvebhya āgamaḥ//padārthīkṛta evānyaiḥ sarvatrābhyupagamyate/saṃbandhas tena śabdārthaḥ pravibhaktuṃ na śakyate* (VP II.3.10–12; translation in Houben: 183–84).

35  MacCracken (2023, p. 6) makes a similar point: 'Relatedness, or relation in its absolute sense, is what orders phenomena into the distinct and unmuddled. Relation is thus not something objectifiable, but rather is indicative of phenomena as having the nature of *vimarśa* (reflective awareness), a core feature of divine subjectivity itself'.

36  While I am inclined to take the Buddhist as the *pūrvapakṣin* in the SS, one might also view the Buddhist as more of an *uttarapakṣa*, given the way in which Utpala appropriates and adopts Dharmakīrti's points rather than directly contradicts them.

37  See, e.g., SP (v. 1): *pāratantryaṃ hi sambandhaḥ siddhe kā paratantratā/tasmāt sarvasya bhāvasya sambandho nāsti tattvataḥ* (cf. also Torella ([1994] 2013, p. 95, f. 21)).

38  *dviṣṭhasyānekarūpatvāt siddhasyānyānapekṣaṇāt pāratantryādyayogāc ca tena kartāpi kalpitaḥ.* The *vṛtti* reads: *sambandho dviṣṭho na caikenātmanobhayatrāvasthitir yuktā na ca dvayoḥ siddhayor anyonyāpekṣātmā nāpi svātmamātraniṣṭhayoḥ pāratantryarūpaḥ sambandhaḥ/ tato yathā jñātṛtvaṃ kalpitaṃ tathā kartṛtvam apīti katham ātmā sarveśara iti?* (cf. translation in Torella ([1994] 2013, pp. 96–97)).

39  See SP 11ab: *dviṣṭho hi kaścit sambandho nāto 'nyat tasya lakṣaṇam.*

40  See PV 40: 'Since all existents essentially abide in their own essence, they partake in the exclusion between [themselves and other] similar and dissimilar things'. *sarve bhāvāḥ svabhāvena svasvabhāva-vyavasthiteḥ/svabhāva-parabhāvābhyāṃ yasmād vyāvṛtti-bhāginaḥ.*

41  Cf. Lawrence (1999, pp. 133–38), who discusses this similarity, and Ratié's recent (2021) study on the ĪPVv, where the nature of action as a form of unity-in-diversity is thoroughly addressed.

42  See SP v 6: *tāmeva cānurundhānaiḥ kriyākārakavācinaḥ/bhāvabhedapratītyarthaṃ saṃyojyante 'bhidhāyakāḥ*

43  *na ca anekasyānekatā-sahabhāvinī ekatā yujyate bhāvābhāva-rūpatvena viruddhatvāt | anekasmād upādānād ekam ekasmād vā anekam anyonyāsaṃsṛṣṭam evātmamātra-paryavasitam udbhavet* (SS: 4).

44  This is just the 'identity principle' of classical Buddhist logic (cf. Eltschinger and Ratié 2013, p. 195).

45  *yathā vijñāna-santatau vyavasthā keṣāṃcit; tatra hi śabda-sparśādi-jñāna-lakṣaṇebhyaḥ samanantara-pratyayebhya ekam aindriyakaṃ vikalpajñānam | ekasmād vā samanantara-pratyayāt pañcāpi śabdādi-viṣayāṇi jāyante | yā punar-ekatā anekatā ca samānāśrayā sā pramāṇabādhitā, bhāvasya vābhāvatā. kevalam-itthaṃ rūpayaiva anayā kalpanāpratītyā sāṃsārika-vyavahāra-nirvartanārtham arthāḥ paraspara-vyāvṛttā api kāryakāraṇarūpatvena avāstavenaiva pratipādyante | rāja-puruṣayor anyonyaṃ svarūpa-viśeṣa-kriyaiva evaṃ nirūpyate sthālyāṃ kāṣṭhair ityādau* ca. 'Accordingly (*yathā*), there is a distinction (*vyavasthā*) of some things (*keṣāṃcit*) in the continuum of awareness. For here the unified sensory cognition is a conceptualization (*vikalpa*) resulting from immediately antecedent conditions (*samanantara-pratyaya*) characterized by awareness of such [sense-fields] as sound and touch; or, sound and all five sense-fields are produced from a single preceding cognition. However, a state of unity and state of multiplicity which share the same locus (*āśraya*), or the non-existence of what is existent, is contradicted by *pramāṇa*s. It's just that objects—though mutually distinguished by such a conceptualization to accomplish conventional life in *saṃsāra*—are bestowed with a form of cause and effect that is entirely unreal. Thus, an action with a specific nature is indicated with respect to the both the king and servant, and likewise "in a cauldron with firewood," etc . . . "' (SS: 4).

46  *ata eva pratītikāla eva sāmānyasyeva saṃbandhasyābhyupagamaḥ* (SS: 4).

47  As Dunne (2004) notes, Dharmakīrti's critique of temporal extension suggests that a mereological analysis of wholes represents the paradigm critique of any and all entities that are 'whole-like: that is, they exhibit 'distribution' (*anvaya*). A whole is a distributed entity in that it is a single real thing that is somehow instantiated in other single real things that are its parts. The same may be held of a perdurant entity that allegedly endures over time: to be real, it must be a single thing distributed over numerous temporal instances' (42).

48  *iha bhāvānāṃ saṃbandho vicārtyate: kastāvat-saṃbandha-śabdārthaḥ saṃsargaḥ saṃparkaḥ saṃśleṣaḥ saṃbandha ityapyukte na vivṛtaḥ saṃbandhārthaḥ pratīyate | kiṃ nairantaryaṃ saṃśleṣaḥ utānyat kiṃcit? nairantaryaṃ cet dūrasthayoḥ pitāputrayoḥ sa na syāt, ayaḥśalākayoḥ saṃnikṛṣṭayor vibhudravyayor api ca syāt* (SS: 1). Note that 'dependence' (*pāratantrya*), 'fusion' (*saṃśleṣa*) and 'requirement' (*apekṣā*) are addressed in the three opening verses of the SP, respectively.

<sup>49</sup> 'Therefore, 'relation' (*saṃbandha*) should be emphatically designated as primary (*mukhya*). In this regard, it is said that these—relation, combination, [and] fusion—are in the first place 'synonyms' (*paryāya*). And that meaning of 'fusion' is regarded as the unity of what is multiple (*aneka*). However, there is neither being only multliple, nor being only unitary; rather this referent, i.e., *saṃbandha*, requires both conditions.' *tasmān muktakaṇṭham eva mukhyaḥ saṃbandho 'bhidhātavya iti | tatrocyate saṃbandhaḥ saṃparkaḥ saṃśleṣa ity ete paryāyās tāvad bhavanti. sa ca saṃśleṣārtho 'nekasyaikatā kathyate, na tv anekataiva nāpi ekataiva api tu ubhayāvasthāpekṣo 'yam arthaḥ saṃbandhaḥ* (SS: 2).

<sup>50</sup> *tatrocyate pratītis tāvad-anekaika-rūpatāyāṃ saṃbandhābhidhānāya bhavadbhir apy abhyupagantaiva yenoktam "ity amiśrāḥ svayaṃ bhāvāstān yojayati kalpanā"* [SP v.5cd] *iti kalpanā-rūpatvaṃ ca nāsyā doṣaḥ | ghaṭo 'yaṃ paṭo 'yam ity api vikalpaḥ kalpanaiva* (SS:4-5).

<sup>51</sup> See ĪPK II.2.1: *kriya-saṃbandha-sāmānya-dravyadik-kāla-buddhayaḥ/satyāḥ sthairyopayogābhyāṃ ekānekāśraya matāḥ.* 'The concepts of action, relation, universal, substance, space and time, which are based on unity-in-diversity, are to be considered real (*satya*) because of their permanence and efficacy'.

<sup>52</sup> *athātra pratyakṣāvabhāso 'pi tathā ghaṭapaṭādirūpa eveti na kalpanātvam* (SS: 5).

<sup>53</sup> *ihāpi rājñaḥ puruṣaḥ sthālyāmodana ity ādāv api pratyakṣāvabhāso; na tatheti kuto 'vagatam? | avaśyam eva ca pratyakṣāvabhāso 'py atra tathaivābhyupagantavyaḥ. abhyupagata eva vā bhavadbhiḥ yenoktam "arthākārapratibhāsa-saṃlagnatvābādhakenāpi duruddharo 'yaṃ bhramaḥ" iti.* 'However, here as well—i.e., in cases such as 'the king's servant' and 'rice in a pot,' etc.—there is also a perceptual appearance; why are [these] not understood accordingly [i.e., as perceptual]? Moreover (*ca*), these perceptual appearances should also necessarily be acknowledged in just this way; or rather (*vā*), it is accepted by you, such that you yourself have said, 'insofar as it is closely connected with the appearance of aspects of objects, this confusion is difficult even for a defeater (*bādhaka*) to uproot [viz., to the extent that the cognition is perceptual]' (SS: 5). Try as I might, I was unable to locate the primary source of Utpala's citation. Assuming it is from Dharmakīrti, it is not from the PV or the PVS (lamentably, Eltschinger (2021, p. 115, f.14) and Allport (230) also apparently could not locate it).

<sup>54</sup> The language of 'internal' and 'external' is, of course, an etic distinction. MacCracken (2023, p. 15) deliberately avoids talk of 'internal' and 'external' relations in the Indian context because it is 'confusing enough between Russell and Bradley without adding what Indian philosophy means by internal and external into the muddle.' While it is true that the use of 'internal' is contested in the Russell/Bradley debate, I still maintain that the dichotomy is an indispensable hermeneutic for dissecting Dharmakīrti and Utpaladeva on relations, not to mention vital for a fruitful cross-cultural analysis. That is, some talk of internal and external relations is necessary to compare the relational realism of Pratyabhijñā with the pragmatism of C.S. Peirce, the radical empiricism of William James, and/or the process philosophy of A. N. Whitehead (my forthcoming dissertation (2024) discusses these issues in depth).

<sup>55</sup> In the ĪPK, Utpala claims that the appearance of cognitive overturning itself presupposes a relation between cognitions established by the synthetic power of the *ātman*: *bādhyabādhakabhāvo 'pi svātmaniṣṭhāvirodhinām/jñānānām udiyāt ekapramātrpariniṣṭhiteḥ.* 'Even the overturning-overturned relation between cognitions, which are self-contained and do not contradict one another, obtains [solely] in virtue of their resting on a single knower' (ĪPK I.7.6; translation, with slight changes, in Torella ([1994] 2013, p. 139)). See Rastogi (1986) and Nemec (2012) for an analysis of Abhinavagupta's theory of error and its relation to Utpaladeva's. (Please see the Appendix A).

<sup>56</sup> *na hi artha-pratibhāsa-saṃlagnatvād bādhakena api tv artha-pratibhāsa-sādṛśya-sadbhāva-mātrāt. rajata-bhrame 'pi śuktikā-sādṛśya-sadbhāvo 'py asty eva, sādṛśya-viṣayā eva hi sarvā bhrāntayaḥ sādṛśya-vyatirekeṇa cānyā artha-pratibhāsa-saṃlagnatayaiva yuktāḥ* (SS: 5).

<sup>57</sup> Cf. Ratié (2011b, 2014) for a discussion on the Pratyabhijñā proof that nothing (including perceptual objects) can appear external to consciousness.

<sup>58</sup> See ĪPK I.6.2–3. For the Pratyabhijñā appropriation of the Buddhist doctrine of *āpoha*, see Prueitt (2016, 2017).

<sup>59</sup> We must view this commitment as motivated by the epistemic doctrine of *intrinsic validity* (*svataḥ prāmāṇya*), in which the property of a cognition's appearing valid is internally related to the appearance of a cognition itself. Thus, all cognitions must be considered valid insofar as they are not practically overturned. See Abhinava's formulation: *satya eva yataḥ sthiro bādhakenānunmūlyamāna-vimarśaḥ saṃvādavāṃś ca abhisaṃhitāyāṃ grāmaprāptilakṣaṇāyāṃ kriyām upayogī.* 'Truth is just that which is (i) permanent, viz., a judgment's not being uprooted by a countervailing cognition, and (ii) manifests (*saṃvāda*) as efficacious in conventional activity characterized by the cognition of many' (ĪPVV III: 29; see also Torella ([1994] 2013, p. 157, f.4)). This definition of cognitions as intrinsically valid until overturned was famously developed by Kumārila (cf. Immerman (2018), Arnold (2001) and Taber (1992)).

<sup>60</sup> Cf. also ĪPK (I.5.13–14): *citiḥ pratyavamarśātmā parāvāk svarasoditā/svātantryam etan mukhyaṃ tad aiśvaryaṃ paramātmanaḥ. sā sphurattā mahāsattā deśakālāviśeṣinī/saiṣa sāratayā proktā hṛdayaṃ parameṣṭhinaḥ* (cf. translation in Torella ([1994] 2013, p. 120)).

<sup>61</sup> *svātmaniṣṭhā viviktābhā bhāvā ekapramātari anyonyānvayarūpaikyayujaḥ saṃbandhadhīpadam. vṛtti: rājñaḥ puruṣa ityādisaṃbandhadhiyo 'ntaḥsamanvayād aikyaṃ bahiḥ saṃbandhibhedaṃ cālambante* (ĪPK(V) II.2.4; translation in Torella ([1994] 2013, p. 159)).

<sup>62</sup> *yadi ekatāmātram eva saṃbandhaḥ syāt, yāvatānekatāṃśād aṃśenaikatā saṃbandhaḥ | tataś ca rāja-puruṣoparaktaikatānyaḥ saṃbandhaḥ pitā-putroparāgopalakṣaṇa-vilakṣaṇa eva* (SS: 6).

<sup>63</sup> Cf. MacCracken (2023, p. 10). Being Is Relating: Continuity-in-Change in the Sambandhasiddhi of Utpaladeva.

<sup>64</sup> In another portion of the text, he equates this same process of grammatical unification to the distinct words (e.g., the 'elephant-horses' that belong to the king) that function like letters of a compound (i.e., *hastyādi-śabdā varṇa-tulyāḥ*): 'For, insofar as it is situated in a single judgement, a word is [itself] singular. Then, due to the application (*adhyāsa*) of a single word, the referent

(*artha*) is also just singular.' *eka-parāmarśa-sthito hi śabda eko bhavati | tad-eka-śabdādhyāsād artho 'py eka eva* (SS:7). There is thus a 'self-similarity' to the 'chunking' capacities of language, where semantic properties always coincide with discerning a holistic unity in a diversity of elements. (For an interesting parallel to this in cognitive science and computation, see Hofstadter (1979, p. 294) on the process of 'chunking' syntactic information).

65    *dvayoś caikye 'pi viśeṣaṇaṃ viśeṣyīkṛta-svarūpaṃ viśeṣyātmanā cakāsti svarūpeṇāpi cāvabhāti 'rājñaḥ puruṣa' iti | 'rājapuruṣa' iti tu viśeṣaṇa-bhūto rājā sarvathā parihārita-svarūpo viśeṣyātmatām evaikāntenāpannaḥ prathate—iti na tatra sambandhavāco yuktiḥ* (SS: 8–9).

66    *nīlam-utpalam ity atrāpi utpalāntaḥ praviṣṭaṃ nīlam iti nīlavad utpalaṃ pradhānam | sthālyāṃ kāṣṭhair ity atrāpi kartrāśritāṃ kriyām upalīnāḥ sthālyādayaḥ prakāśante | 'ghaṭasyābhāva' ity atrāpi abhāvo vikalpabuddhāv antarnīta-ghaṭaḥ prādhānyenāvabhāti | 'ayamasmādanya' ity anyārtho 'nyatvāparityāgena ivāntarnītāparāny-ārtho viśeṣya iti | evaṃ sarvatrānumantavyam.* 'In the case of "blue lotus" as well, the blue is subsumed within (*praviṣṭa*) the lotus, and thus the blossom *that possesses blueness* is the principal member (*pradhāna*). Here also with respect to "in the pot with firewood", things such as pots, insofar as they are absorbed into (*upalīnāḥ*) an action, appear as dependent upon the agent. In the case of "the absence of a pot" as well, an absence, in which a pot is included within the conceptualization, appears as primary. "This is other than this"—in this case, without at all (*eva*) relinquishing the sense of otherness, the meaning of 'other' is 'something (*artha*) different from another that includes [the former],' [which is the thing] to be characterized. This should be acknowledged in all cases' (SS: 9).

67    This is why Utpala affirms that, even though a sequence of objects is determined by the conventional subject, 'there is no sequence of understanding [i.e., with respect to the disclosure of Śiva]; just the single manifestation of an object at just one moment— precisely *this* discloses (*āviṣkaroti*) the nature (*svarūpam*) that consists only in the fact that Śiva is relation (*sambandhaśivatā*).' *na ca saṃvidaḥ kramo 'sti ekaivaikatraiva kṣaṇe 'rtha-prakāśanā, saiva sambandha-śivatā-mayam eva svarūpam āviṣkaroti* (SS: 9) (cf. MacCracken 2023, p. 13).

68    On the nature of the *agential* unity of the *kārakas* in the Pratyabhijñā system, see Lawrence (1998): '[A]ll of the *kārakas* are understood to function in accomplishing the overall action or process (*vyāpāra*) expressed by the verb. They do this through their own *subordinate processes.* The pan *holds* the rice, the fire *heats* it, and so forth. Where are all the subordinate processes synthesized into the larger one? This is understood to be accomplished by the agent, who is the locus of the overall process (*vyāpārāśraya*)' (597).

69    Note that these relations must be 'figurative' because the insentient nature of particular objects is not self-illuminating, and thus they cannot literally instantiate the relational properties of 'dependence' or 'requirement,' which require the synthetic perspective of an 'intentional' level of description. On this point, see ĪPK (II.4.14–15): *asmin satīdam astīti kāryakāraṇatāpi yā/sāpy apekṣāvihīnānāṃ jāḍānāṃ nopapadyate. na hi svātmaikaniṣṭhānām anusandhānavarjinām/sadasattāpade 'py eṣa saptamyarthaḥ prakalpyate.* 'The relation of cause and effect as well, i.e., 'when this exists, this comes to be,' is not admissible for realities that are insentient and as such incapable of 'requiring.' In fact, the meaning of the locative case [i.e., 'when this exists'] may not be applied to self-contained entities, incapable of intentional synthesis (*anusaṃdhāna*) whether [cause and effect] are considered existent or non-existent' (translation, with slight adjustments, in Torella ([1994] 2013, pp. 183–84)). (See Bronner 2016 (95) for the semantics of *guṇavṛtti* and its development in the context of Kashmiri poetics).

70    Cf. SS(6): 'Furthermore, in these terms, sometimes the referent of relation could possess a secondary sense (*gauṇavṛtti*) because it refers to an inferior aspect (*aparabhāga*) of the inferior-superior state (*parāpara*). And (*ca*) this [relation] of two terms is distinguished simply as assister and assisted, and this must necessarily occur (*pratipādanīya*); if this [relation] does not occur, then a determination (*pratīti*) of these two sequential [relata] (i.e., assister and assisted) *as simultaneous* for the sake of acquisition and relinquishment could not otherwise be effective (*ghaṭeta*)—and therefore there would be a disruption (*lopa*) of conventional life.' *tatrāpi kadācit parāparadaśāyām aparabhāgāpekṣaṇād gauṇavṛttyā sambandhārthaḥ sambhavet, upakāryopakārakayor eva ca viśiṣṭayoḥ sambandhaḥ viśiṣṭarūpaḥ, sa ca pratipādanīyo 'vaśyam eva; tad apratipādane tayor upakāryopakārakayoḥ kramikayor yaugapadyena pratītir hānopādānārtham anyathā na ghaṭeta, tataś ca vyavahāra-lopaḥ syāt.*

71    See MacCracken (2023, p. 6), Ratié (2021, pp. 95–96) and Torella ([1994] 2013, p. 125, f.42) on this point. Note that despite his general repudiation of Bhartṛhari (cf. Nemec 2011), the example of running and recollection is also used by Somānanda in his ŚD (1.9–11ab).

72    'For even with respect to one moving along a path (*mārgagati*), appearances such as those consisting of the sensations of things— e.g., like the grass (*tṛna*) situated (*vartin*) on the side (*pārśva*) [of the path]—are not admitted as existents (*sattvena*) apart from judgment, due to not being remembered (*smaryamāṇa*) [later]. Neither, in that case (*tadā*), does it make sense to establish the existence of those entities (*sattā*) by inference (from the presence of the collection of sense organs such as sight), because of the absence of the attention of mind. When that is present [i.e., attention], there will necessarily be a judgment at that time (*tadānīṃ*) of such things as "grass" etc., and that is now a memory.' *na hi mārgagatipravṛttasyāpi pārśvavartitṛṇādivastusparśarūpādipratibhāsāḥ parāmarśarahitāḥ sattvenābhyupagantuṃ pāryante smaryamāṇatvābhāvāt | nāpi teṣāṃ tadā cakṣurādikāraṇasāmagrīsadbhāvenānumānasiddhā sattā yujyate manovadhānābhāvāt | tadbhāve'vaśyaṃbhāvī tadānīṃ tṛṇādiparāmarśa idānīṃ ca smaraṇam* (SS:9).

73    See ĪPK I.5.19: 'Even at the moment of the direct perception there is a reflective awareness. How otherwise could one account for such actions as running and so on, if they were thought of as being devoid of determinate awareness?' *sākṣātkārakṣaṇe 'py asti vimarśaḥ katham anyathā//dhāvanādy upapadyeta pratisaṃdhānavarjitam?* (translation in Torella [1994] 2013, pp. 125–26). MacCracken (2023, p. 6) likewise notes that '[i]ncipient interpretability is present and real even in what is common-sensically called perception, that is, perceptual as opposed to conceptual cognition'.

74  Most notably in this regard, see Peirce (1871), where he claims that the medieval opposition between *realism* and *nominalism* partly inspires misguided subjectivist strains of modern '*idealism*' (i.e., chiefly exemplified in the nominalism of Berkeley's idealistic philosophy). I hope to say more about this Peircean observation in future publications on the 'objective' idealism of Pratyabhijñā as a corrective to the epistemic idealism of Dharmakīrtian Sautrāntika Buddhism.

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
