# Peer review of "Sambandha as a ‘Śakti-of-Śaktis’: Bhartṛhari’s Influence on the Relational Realism of Pratyabhijñā"

_religions, doi:10.3390/rel14070836_

Round 1

Reviewer 1 Report

This is a very substantial paper, well articulated and documented. It should be accepted as it is.

I've only detected two minor typos:

p4 "pre-verbalized from" should be "pre-verbalized form" 

p14 "āpoha" should be "apoha"

Author Response

I have corrected the two noted typos. Thank you for the input.

Reviewer 2 Report

I have read the article with great interest and believe it delivers on its promises. The main argument presented is that the Pratyabhijñā Śaivas have adopted a transcendental understanding of relation, drawing from Bhartṛhari's treatment of the same notion in the VP. The Pratyabhijñā then utilises this interpretation of relation to counter the Buddhist scepticism regarding the existence of this notion. The Pratyabhijñā asserts that the idea of relation is inherently embedded in all synthetic judgments, even those put forward by Buddhists, such as the claim that "There are no relations but only discrete, momentary, and partless particulars."

The paper possesses several merits:

  • The philosophical discussion of the issue is lucid, refined, and effectively targeted towards a philosophical audience. It is commendable that philosophers can now appreciate a counterargument to Buddhist nominalism, which is generally a more familiar standpoint to them.
  • Moreover, the author grounds her discussion in the texts, particularly Utpaladeva's Saṃbandhasiddhi (SS), thus demonstrating direct familiarity with the original sources.
  • Utpaladeva's SS is an understudied text, and this paper sheds new light on a compelling and challenging work.
  • The paper also highlights another aspect of the subtle incorporation of Bhartṛhari's ideas by the Śaiva thinkers. Although Bhartṛhari is not explicitly cited as a source of inspiration by Utpaladeva concerning the notion of saṃbandha, the author's overall reconstruction is reasonable, also considering what is already known about the influence of the grammarian on the Pratyabhijñā milieu.

I would like to offer a few minor suggestions:

  • To gain further insight into Pratyabhijñā's semantics (especially on how abhidhā works), the author could consult Ferrante's "The Place of Language in the Philosophy of Recognition" (2021) or Torella's "The Word in Abhinavagupta Bṛhat-Vimarśinī. (2001).
  • Regarding pratibhāexploring the recent contributions of Hugo David and Nilanjan Das is certainly worthwhile.
  • Some typos are scattered throughout the text: in line 195, the author may mean Śsam instead of VP; in footnote 54, it should be Torella instead of Torrella; Biardeau is misspelt once, and so on. Please check carefully!
  • For a textual reference on the progressive manifestation of sphoṭa, the author can refer to VP 1.84-86 with the accompanying svavṛtti.

Author Response

I have amended all the typos, and included the suggested referents in relevant footnotes—8, 16, and 13, respectively.

Thank you for your feedback!